# Bridges over Troubled Waters? The Political Economy of Public-Private Partnerships in the Water Sector

George Shambaugh [1,2,*] and Shareen Joshi [1]

1   Edmund Walsh School of Foreign Service, Georgetown University, Washington, DC 20057, USA; sj244@georgetown.edu
2   Department of Government, Georgetown University, Washington, DC 20057, USA
*   Correspondence: George.Shambaugh@georgetown.edu; Tel.: +1-202-687-2979

**Abstract:** Global concerns about water security and water scarcity are motivating local governments, investors, and international financial institutions to prioritize investments in the water sector. Over the past thirty years, public–private partnerships (PPPs) have been popular mechanisms for encouraging private sector investment and helping local governments overcome economic, political, and technical challenges associated with large infrastructure projects in the water, electricity, and transportation sectors. We argue that the political economy factors that affect the prevalence of PPPs in the water sector—which must serve broad populations of people at low cost—are different than other types of infrastructure projects. We use the World Bank's Private Participation in Infrastructure (PPI) database to explore factors that affect the likelihood that PPPs will be initiated in water relative to other sectors, and in water treatment relative to water utilities. We demonstrate that the likelihoods of PPPs in the water sector and water treatment are positively correlated with levels of output from industries that are water-intensive and pollution-intensive when the host country relies heavily on fossil fuels to generate electricity. Furthermore, when corruption levels are high, projects are more likely to be initiated in water than in other sectors, but those investments are more likely to be in water utilities than water treatment.

**Keywords:** water; infrastructure; public-private partnerships

## 1. Introduction

Access to clean water and sanitation, fundamental to human life, have been recognized by the United Nations as essential for the realization of all other human rights [1] Yet almost 1 billion people lack access to safe drinking water and 2.2 billion people—nearly fifty percent of the developing world's population—lack adequate sanitation facilities [2]. In recognition of this precarious situation, the Sustainable Development Goals aim to achieve universal and equitable access to safe and affordable drinking water, sanitation, and hygiene for all by 2030 and also "increase the efficiency of water-use across all sectors" by this time. Building on this, the World Bank has set a goal of building a "water-secure future for all" by improving funding, efficiency, and financial sustainability of the water sector through investment in technology and expertise [3].

Commitments to water security from international organizations often manifest themselves in support for large-scale infrastructure projects. These typically either focus on the supply and distribution of potable water or the management of wastewater. In recent years, there has been considerable emphasis on the use of public–private partnerships (PPPs) in these projects to encourage private sector investment and help overcome economic, political, and technical challenges associated with such projects [4]. PPPs take multiple forms but are generally characterized by long-term contracts between public and private partners, where the private partner usually designs, finances, builds, and operates the infrastructure or the service [5].

Estimates from the World Bank suggests that the number of PPPs in the water sector grew steadily through the 1990s, peaked around 2007, and fluctuate around a lower level thereafter (Figures 1 and 2). The drop and lack of resurgence in PPPs are often attributed to project-specific failures [6,7], country-specific policies [8], or challenges of conducting business in politically fragile and economically weak states [9]. Commonly reported issues include overbuilding, underbidding, inaccurate cost estimation for capital as well as maintenance, cost-saving decisions by private partners, overreliance on debt financing, lack of expertise in management, poor service provision, poor customer service, and an overall public skepticism of privatization compounded by financial uncertainty [7,10].

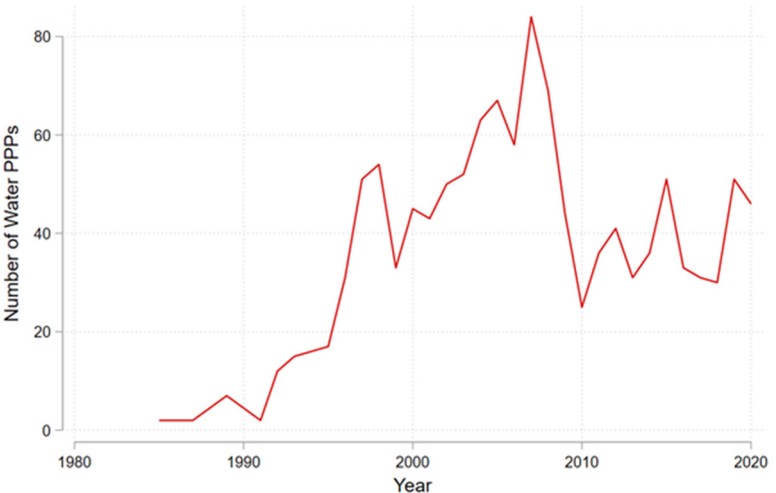

**Figure 1.** Total number of PPPs, 1980–2021. Source: World Bank, PPI Database (2021).

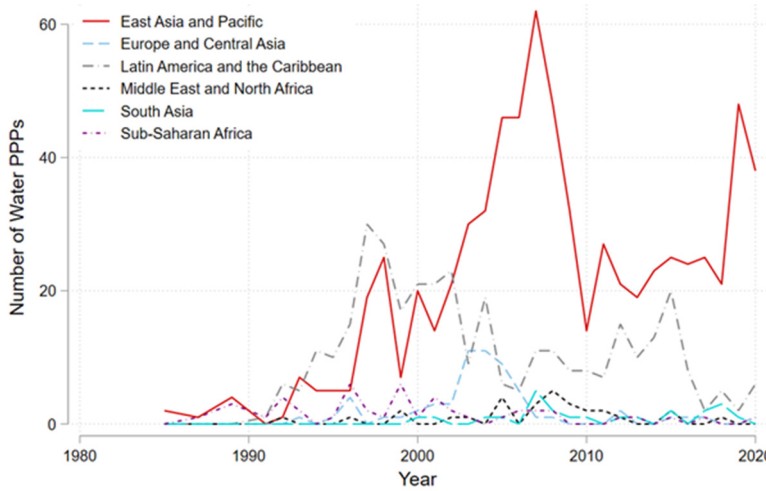

**Figure 2.** Total number of PPPs, 1980–2021. Source: World Bank, PPI Database (2021).

While research on the failures of water PPPs provides valuable perspectives regarding project sustainability, many questions remain about the underlying motivations for funding infrastructure projects and the conditions under which the PPPs and World Bank support for PPPs in the water sector are likely. Policy experts acknowledge that infrastructure development in the water sector involve a set of incentives and challenges that make it different from other sectors [11]. At a minimum, water is unusual for having the dual characteristics of a basic human need and an environmental public good [12]. There are also unique societal and political pressures on local governments to provide access to potable water and manage water pollution at reasonable cost to entire populations, particularly in the presence of toxic industries [13,14]. Limited profitability, the prevalence

of national monopolies, and the likelihood of corruption in this sector compound these factors in making the water sector less attractive than others to commercial involvement [15]. Challenges associated with water projects are likely to be more severe in settings where rapid economic growth and industrialization has increased demands for clean water and generated higher levels of water pollution [10]. We posit that the combination of these local needs and challenges increase incentives for local governments to seek external financial support and private sector partners to help them overcome market, political, and technical challenges in the water sector. External funding obtained via PPPs and international financial institutions like the World Bank could, in essence, provide a bridge over troubled waters. This paper explores this possibility by analyzing the conditions under which national governments and private sector actors are likely to engage in water sector projects.

We argue that the involvement of public–private partnerships in infrastructure projects is best understood within a political economy framework that considers sector-specific variations in the political and economic motivations for investment. We demonstrate that investment patterns in the water sector differ from those in other infrastructure sectors. Previous work has highlighted the salience of these political factors in driving investments in the water sector [16]. Our work takes this further by focusing on the subsectors where the funds are directed. Despite the common physical connection between the provision of potable water and the mitigation of water pollution, factors that affect investment in water utilities vary significantly than those that affect water treatment.

Our analysis relies on the World Bank's Private Participation in Infrastructure (PPI) database. This database is used by the World Bank to identify and disseminate information on private participation in infrastructure projects in low- and middle-income countries. This dataset has been previously used to analyze the funding of PPP projects [17], but those analyses have not focused on the sectors or sub-sectors for the funding decisions.

We first compare the likelihood of PPPs in the water sector compared to others in the PPI database. Next, we look at the breakdown of investments within the water sector. We demonstrate that PPPs in water treatment are likely under different conditions than those associated with water provision, power, or transportation. In most developing countries, the provision of clean water, electricity, and roads are recognized as necessary conditions for commercial as well as social development. They are often associated with higher levels of productivity, trade, and foreign direct investment. In contrast, investment in the management and treatment of water pollution tends to lag these factors and is driven primarily by the level of pollution they generate. These differences persist despite variation in national factors such as the representativeness and effectiveness of a country's political system and the level of inequality, and international factors including net foreign direct investment, export volume, and changes in global demand for exports in highly polluting sectors. From an environmental perspective, it is good and seemingly obvious to confirm that the likelihood of investment in water cleanup is positively correlated output from dirty industries like textiles and manufacturing, and the extent that a country relies on fossil fuels to generate power. At the same time, it is important and somewhat troubling to recognize that this effect is sector-specific and that the likelihood of PPPs in the electricity sector are not affected by manufacturing output or fossil fuel reliance, and the likelihood of PPPs in transportation are not affected by manufacturing output, textile output, or reliance on fossil fuels.

In a nutshell, investment in the water sector is associated with different factors than other infrastructure investments, and investment in water treatment PPPs is distinct from and more likely to be motivated by industrial pollution than the provision of potable water even though potable water supplies are often stressed by private sector activity.

## 2. Materials and Methods

This paper focuses on the political economy drivers of public–private partnerships in the water sector. Our first hypothesis is that likelihood of projects and PPPs in the water

sector is driven by the size of water-consuming and polluting industries. A vast interdisciplinary literature shaped by the voices of scientists, academics, bureaucrats, industry representatives, and citizens has underscored the importance of water governance [18–20]. While there is little consensus on a single governance framework, there is a near-complete consensus that the initiation of PPPs in the water sector are triggered by different factors than investments in other sectors. At a minimum, while infrastructure projects in water, transportation, and electricity all generally require large, sustained investments in infrastructure and are necessary for economic development, water is a non-substitutable basic human need [19–21]. Consequently, debates over the availability, access, and pricing of water are almost always more intense, more politicized, and more scrutinized than other forms of infrastructure. Attracting private sector engagement in the water sector is also likely to be more difficult than in transportation or electricity because the economic rate of return for investments in the water sector, and water cleanup in particular, is likely to be lower and more difficult to sustain [15].

Our second hypothesis is that investments in water provision are likely to be driven by a different set of factors that water cleanup. We posit that relatively low political and economic rewards associated with large public expenditures in water treatment reduce the political incentives maintain public monopolies that are common among water utilities. The lack political and economic rewards also decrease the incentives for national governments to expend scarce public funding on large water treatment projects. To overcome these political and market failures, local governments often seek external financing and private sector support for water treatment projects. This provides an opportunity for entities like the World Bank that are interested in promoting private sector engagement in underserved arenas.

Increasing in economic activity are often associated with both greater stress on potable water supplies and higher levels of water pollution. Recent studies demonstrate, for example, that higher levels of trade in water-intensive or polluting sectors like manufacturing, textiles, and agriculture have a negative impact on access to potable water [18] and water pollution [22,23]. These effects have been shown to generate political pressure on national governments to improve access to potable water in countries with lower levels of inequality and those where people in the middle and lower classes are better able to organize politically [16]. This suggests that government support for PPPs involving potable water should be more numerous in countries that have large polluting sectors (textiles, manufacturing, and agriculture), lower levels of inequality, and greater opportunities for democratic participation.

While similar political dynamics may pertain in the water treatment sector, we posit that the political rewards of water treatment projects are lower than those associated with their provision of potable water or other infrastructure projects. Furthermore, the positive commercial spillover effects of infrastructure development in pollution cleanup are less direct and immediate than those associated with investments in water utilities, roads, and electricity. The reduced scope and immediacy of spillover effects suggests that motivations for investing in water treatment efforts will be narrower than in other sectors. Thus, specific drivers, like the level of water pollution generated by particular industries, are likely to have greater effects on investments in the water treatment relative to other sectors.

We test both hypotheses in a simple econometric framework. The main dependent variables are the incidence of PPPs in the water sector or the water treatment subsector. The primary explanatory variables include the level of output in polluting industries (particularly textiles and manufacturing) and proportion of electricity generated by fossil fuels. Political control variables include the level of democratization and corruption in the host country. Economic control variables include net foreign direct investment, trade exports, global prices for exports in the textile industry, and measures of inequality and poverty. We use several sources of data in our analysis. These are described below.

Data on Public–Private Partnerships: Our analysis relies on the World Bank's Private Participation in Infrastructure Projects Database (PPI). The database highlights the con-

tractual arrangements used to attract private investment, the sources and destination of investment flows, and information about the principal investors.

We focus on the probability that PPPs will be created in different sectors in middle- and low-income countries around the world in a given year. The PPI contains information about 10,427 infrastructure projects dating from 1984 to 2020. It has over 50 fields per project record. These include host country, financial closure year, infrastructure services provided, type of private participation, technology, capacity, project location, private sponsors, private debt providers, and development bank support. The PPI includes 5898 projects in the power sector (57 percent), 3017 projects in the transportation sector (29 percent), and 1512 in the water sector (14 percent). Of those in the water sector, 498 are water utility projects (33 percent) and 1014 are water treatment projects (67 percent). In total, 8709 (84 percent) of the projects in the data set are PPPs. Almost all water treatment projects are PPPs (675 out of 682, or 99 percent), and almost all water utilities are PPPs (472 out of 498, or 95 percent).

The PPI represents the best efforts of a research team at World Bank to compile publicly available information on those projects, and it is not intended to be a fully comprehensive resource. Some projects—particularly those involving local and small-scale operators— tend to be omitted because they are usually not reported by major news sources, databases, government websites, and other sources that used the PPI.

We organize the PPI data at the project-level on a country-year basis. The country refers to the country where the investment takes place. The project-year designation identifies the first year the project takes place and enables us to identify the number of active projects in a country in any given year. All regressions include country-specific and year-specific fixed effects. Our main dependent variables are binary and are analyzed using logit regressions. We analyze factors that affect the likelihood that projects will be initiated in the water sector relative to other sectors, and the likelihood that they will take place in the water treatment subsector.

Productivity by Sector and Pollution: We posit that the likelihood of PPPs in the water sector will depend on the aggregate production level in industries that consume large amounts of water and generate high levels of water pollution. The textile and manufacturing industries do both. They are large water consumers and their wastewater often contains many types of pathogens, oxygen-demanding substances, and inorganic and synthetic organic chemicals. In many developing countries across the world, this wastewater is often discharged directly into water streams without proper treatment [10,14]. Therefore, we approximate the level of water demand and pollution using aggregate production level in the textiles and manufacturing industries. Production levels are measured in terms of the value added of output the textile and manufacturing sectors in the host country in billions of constant U.S. dollars (2017). To further capture the level of environmental pressure, we also consider the extent of the country's electrical power that is generated from fossil fuels. To measure the intensity of pollution in the textile and manufacturing industries we interact the two variables (textile production × electricity from fossil fuels, and manufacturing production x electricity from fossil fuels).

Governance and Corruption Indicators: We estimate the impact of political regime type using the POLITY 2 composite score of democratization. The democracy score is a composite of 4 dimensions of governance including the competitiveness of executive recruitment, openness of executive recruitment, constraints on the executive, and the competitiveness of political participation. It ranges from −10 (hereditary monarchy) to +10 (consolidated democracy). Countries with scores from −10 to −6 are considered autocracies, those with scores of +6 to + 10 are considered democracies. We estimate corruption using the World Bank Worldwide Governance Indicator (WGI). The WGI is a composite of 6 dimensions of governance including voice and accountability, political stability and absence of violence, government effectiveness, regulatory quality, rule of law, and control of corruption. In our dataset, the WGI indicator ranges from 0 (no corruption) to 5 (high corruption). The WGI is highly correlated with other commonly used indicators of corruption including Transparency International's Corruption Perception Index (CPI)

(r = 0.756 **). Sensitivity tests show no significant differences in the results in when the CPI is used in place of the is used in place of the WGI.

Economic exposure and inequality: We measure economic exposure in terms of net FDI inflows and net exports in billions of US dollars. We estimate inequality using the poverty headcount ration at 1.90 U.S. dollars per day. These data are gathered from the World Bank's World Development Indicators database. The poverty headcount ratio based on USD 1.90 per day is highly correlated with the LMIC poverty rate based on 3.20 per day (r = 0.9419 **) and the UMIC rate based on USD 5.50 per day (r = 0.8555 **) in our data set. The results remain consistent regardless of which indicator is used. GDP is excluded because the regressions include country-specific effects which are highly correlated with GDP.

We approximate the impact of global demand for the products generated in the textile and manufacturing industries using the average global prices for cotton, leather, and synthetic fibers over the previous five years. We chose five years to account for the time it would take investors to reach an investment agreement following a price rise. The results remain consistent if the average price is estimated over the past one year instead of the past five. We obtained these data from the International Monetary Fund.

A full set of summary statistics for all key dependent and independent variables are presented in Table 1.

**Table 1.** Summary Statistics.

| Variable Name | N | Mean | SD | Min | Max |
|---|---|---|---|---|---|
| **Project Details** | | | | | |
| PPP Project | 10,427 | 0.835 | 0.371 | 0.000 | 1.000 |
| PPP Project-Electricity Sector | 10,427 | 0.395 | 0.489 | 0.000 | 1.000 |
| PPP Project-Transport Sector | 10,427 | 0.218 | 0.413 | 0.000 | 1.000 |
| PPP Project-Water Sector | 10,427 | 0.137 | 0.344 | 0.000 | 1.000 |
| PPP Project-Water Sector-Water Treatment | 10,427 | 0.091 | 0.288 | 0.000 | 1.000 |
| PPP Project-Water Sector-Water Utility | 10,427 | 0.045 | 0.208 | 0.000 | 1.000 |
| **Economic Activity** | | | | | |
| Manufacturing value-added in billions of USD (2017) | 9879 | 424.453 | 933.567 | 0.000 | 3868.458 |
| Net FDI inflows in billions of USD (2017) | 10,392 | 40.398 | 65.360 | −7.574 | 290.928 |
| Textile value-added in billions of USD (2017) | 8228 | 41.689 | 91.042 | 0.000 | 386.533 |
| Proportion of Electricity generated from fossil fuels | 8639 | 0.561 | 0.307 | 0.000 | 1.000 |
| Manufacturing Value Added | 9264 | 17.941 | 6.926 | 0.733 | 50.037 |
| Net FDI inflows in billions of USD (2017) | 10,392 | 40.398 | 65.360 | −7.574 | 290.928 |
| Net Exports in billions of USD (2017) | 10,351 | 368.95 | 688.015 | 0.00805 | 2655.592 |
| **Governance and Corruption** | | | | | |
| Level of Democratization (Polity 2) | 9508 | 3.798 | 6.190 | −9.000 | 10.000 |
| WGI: 0 (low corruption) to 5 (high corruption) | 6912 | 2.885 | 0.352 | 1.271 | 4.222 |
| **Inequality and Poverty** | | | | | |
| Poverty headcount ratio at USD 1.90/day (2011 PPP) | 5634 | 9.811 | 11.026 | 0.000 | 86.200 |
| Poverty headcount ratio at USD 3.20/day (2011 PPP) | 5634 | 22.090 | 19.171 | 0.100 | 96.300 |
| **Prices** | | | | | |
| Avg Cotton price over previous 60 months | 10,427 | 124.901 | 15.593 | 100.445 | 153.652 |
| Avg Leather price over previous 60 months | 10,427 | 210.233 | 32.185 | 172.045 | 292.892 |
| Avg Synthetic fiber price over previous 60 months | 10,427 | 138.561 | 16.834 | 111.162 | 167.117 |

## 3. Results

Our results are presented in Tables 2–5. Tables 2 and 3 analyze factors that affect the likelihood that investment will take place in the water sector relative to other sectors. The dependent variable in estimates 1 through 6 in Tables 2 and 3 takes on the value of 1 for projects in the water sector and 0 for projects in the electricity or transportation sectors. The dependent variable in estimates 7 and 8 in Tables 2 and 3 takes on value of 1 for projects in the electricity and transportation sector, respectively, and 0 for projects in other sectors. The

dependent variable in estimates 1 through 5 in Tables 4 and 5 takes on the value of 1 for water treatment projects and 0 for projects in all other sectors. The dependent variable in estimates 6 through 10 in Tables 4 and 5 takes on the value of 1 for water treatment projects and 0 for water utility projects.

**Table 2.** Sector Logit, Textiles Industry.

| | (1) | (2) | (3) | (4) | (5) | (6) | (7) | (8) |
|---|---|---|---|---|---|---|---|---|
| | **Water Project (Any Type)** | | | | | | **Electricity** | **Transport** |
| **Productivity by Sector** | | | | | | | | |
| Textile value-added, billions of USD | −0.049 *** | −0.034 ** | −0.037 ** | −0.044 ** | −0.042 ** | −0.022 | −0.028 ** | 0.062 *** |
| | (0.012) | (0.013) | (0.013) | (0.013) | (0.013) | (0.014) | (0.010) | (0.015) |
| Proportion of electricity from fossil fuels | 3.026 * | 0.435 | 0.013 | −0.676 | −0.696 | −0.175 | 0.449 | 3.395 ** |
| | (1.211) | (1.456) | (1.554) | (1.596) | (1.599) | (1.659) | (1.073) | (1.297) |
| Textiles × Electricity from fossil fuels | 0.072 *** | 0.050 ** | 0.040 * | 0.049 ** | 0.046 ** | 0.013 | 0.044 ** | −0.052 ** |
| | (0.015) | (0.017) | (0.017) | (0.017) | (0.017) | (0.018) | (0.014) | (0.020) |
| **Governance Indicators** | | | | | | | | |
| Democracy Score (Polity 2) | | −0.017 | −0.011 | 0.034 | 0.033 | 0.062 | −0.029 | 0.049 |
| | | (0.032) | (0.032) | (0.036) | (0.036) | (0.039) | (0.029) | (0.039) |
| Corruption (WGI) | | 1.337 ** | 1.135 * | 0.984 * | 1.008 * | 1.525 ** | −2.039 *** | 0.226 |
| | | (0.442) | (0.452) | (0.456) | (0.457) | (0.494) | (0.363) | (0.392) |
| **Economic Indicators** | | | | | | | | |
| Net FDI inflows in billions of USD (2017) | | | 0.006 | 0.006 | 0.007 | 0.012 ** | 0.000 | −0.012 *** |
| | | | (0.004) | (0.004) | (0.004) | (0.004) | (0.003) | (0.003) |
| Net exports in billions of USD (2017) | | | 0.001 | 0.001 | 0.001 | 0.002 | −0.000 | −0.003 ** |
| | | | (0.001) | (0.001) | (0.001) | (0.001) | (0.001) | (0.001) |
| Poverty HC ratio USD 1.90/day (2011 PPP) | | | | 0.033 ** | 0.033 ** | 0.043 *** | 0.006 | −0.033 *** |
| | | | | (0.011) | (0.011) | (0.012) | (0.008) | (0.009) |
| **Global Demand for Textiles** | | | | | | | | |
| Avg Cotton price, previous 60 months | | | | | −0.335 ** | −0.333 ** | 0.017 | 0.285 ** |
| | | | | | (0.109) | (0.113) | (0.086) | (0.097) |
| Avg Leather price, previous 60 months | | | | | 0.001 | 0.001 | −0.010 | 0.009 |
| | | | | | (0.013) | (0.013) | (0.011) | (0.013) |
| Avg Synthetic fiber price, previous 60 months | | | | | 0.283 * | 0.276 * | 0.122 | −0.418 *** |
| | | | | | (0.129) | (0.134) | (0.104) | (0.115) |
| Chi-squared statistic | 315.052 | 163.752 | 170.614 | 180.452 | 192.548 | 206.657 | 424.679 | 367.778 |
| N | 6847 | 5036 | 5034 | 5034 | 5034 | 4483 | 4638 | 4627 |
| Textile value-added + [Textiles × Electricity] | 0.0234 *** | 0.0164 *** | 0.0024 | 0.0048 | 0.0042 * | −0.0088 | 0.0162 ** | 0.0102 |
| Std. error | 0.0036 | 0.0040 | 0.0075 | 0.0076 | 0.0023 | 0.0078 | 0.0068 | 0.0082 |
| Country Fixed-Effects | Yes | Yes | Yes | Yes | Yes | Yes | Yes | Yes |
| Year Fixed-Effects | Yes | Yes | Yes | Yes | Yes | Yes | Yes | Yes |
| Sample | All Projects | All Projects | All Projects | All Projects | All Projects | Only PPPs | Only PPPs | Only PPPs |

Notes: The full sample has 10,427 projects. All logistic regressions include a full set of country and year fixed-effects. The symbols *, **, and *** indicate 10%, 5% and 1% levels of statistical significance, respectively.

**Table 3.** Sector Logit, All Manufacturing.

| | (1) | (2) | (3) | (4) | (5) | (6) | (7) | (8) |
|---|---|---|---|---|---|---|---|---|
| | Water Project (Any Type) | | | | | | Electricity | Transport |
| **Productivity by Sector** | | | | | | | | |
| Manufacturing value-added in billions of USD | 0.228 *** | 0.231 *** | 0.170 ** | 0.231 *** | 0.225 *** | 0.197 ** | −0.093 | −0.201 ** |
| | (0.030) | (0.036) | (0.052) | (0.055) | (0.055) | (0.061) | (0.064) | (0.063) |
| Proportion of electricity from fossil fuels | 0.043 | 4.795 ** | 4.795 ** | 5.067 ** | 4.847 * | 8.529 ** | −0.825 | −7.668 * |
| | (1.509) | (1.759) | (1.849) | (1.886) | (1.898) | (2.678) | (2.852) | (3.231) |
| Manufacturing × Electricity from fossil fuels | 0.156 * | −0.081 | −0.084 | −0.111 | −0.101 | −0.231 * | 0.058 | 0.397 ** |
| | (0.062) | (0.068) | (0.070) | (0.071) | (0.072) | (0.109) | (0.123) | (0.141) |
| **Governance Indicators** | | | | | | | | |
| Democracy Score (Polity2) | | −0.052 | −0.039 | 0.015 | 0.014 | 0.048 | 0.012 | 0.022 |
| | | (0.028) | (0.029) | (0.033) | (0.033) | (0.036) | (0.024) | (0.031) |
| Corruption (WGI) | | 0.426 | 0.640 | 0.429 | 0.421 | 0.930 * | −1.583 *** | −0.073 |
| | | (0.369) | (0.394) | (0.399) | (0.400) | (0.428) | (0.301) | (0.322) |
| **Economic Indicators** | | | | | | | | |
| Net FDI inflows in billions of USD (2017) | | | 0.006 | 0.007 | 0.007 | 0.010 * | 0.001 | −0.007 * |
| | | | (0.004) | (0.004) | (0.004) | (0.004) | (0.003) | (0.003) |
| Net exports in billions of USD (2017) | | | −0.000 | −0.000 | −0.000 | 0.000 | 0.001 | 0.000 |
| | | | (0.000) | (0.000) | (0.000) | (0.000) | (0.000) | (0.000) |
| Poverty HC ratio USD 1.90/day (2011 PPP) | | | | 0.041 *** | 0.042 *** | 0.049 *** | 0.007 | −0.017 * |
| | | | | (0.011) | (0.011) | (0.012) | (0.007) | (0.008) |
| **Global Demand for Textiles** | | | | | | | | |
| Avg Cotton price, prev 60 months | | | | | −0.376 *** | −0.358 ** | 0.014 | 0.258 ** |
| | | | | | (0.106) | (0.110) | (0.082) | (0.093) |
| Avg Leather price, prev 60 months | | | | | 0.004 | 0.001 | −0.007 | 0.005 |
| | | | | | (0.013) | (0.013) | (0.010) | (0.012) |
| Avg Synthetic fiber price, prev 60 months | | | | | 0.321 ** | 0.300 * | 0.109 | −0.376 *** |
| | | | | | (0.124) | (0.129) | (0.100) | (0.109) |
| Chi-squared Statistic | 375.028 | 172.830 | 175.072 | 191.235 | 206.491 | 216.797 | 449.528 | 326.691 |
| N | 8076 | 5524 | 5510 | 5510 | 5510 | 4806 | 5120 | 5121 |
| Manu value-added + [Manu × Electricity] | 1.469 *** | 1.161 ** | 1.091 | 1.128 * | 1.132 * | 0.966 ** | 0.966 | 1.216 |
| Std. error | 0.084 | 0.072 | 0.078 | 0.082 | 0.083 | 0.094 | 0.102 | 0.146 |
| Country Fixed-Effects | Yes | Yes | Yes | Yes | Yes | Yes | Yes | Yes |
| Year Fixed-Effects | Yes | Yes | Yes | Yes | Yes | Yes | Yes | Yes |
| Sample | All Projects | All Projects | All Projects | All Projects | All Projects | Only PPPs | Only PPPs | Only PPPs |

Notes: Regression analysis of types of PPPs. The full sample has 10,427 projects. All regressions include a full set of country and year fixed-effects. The symbols *, **, and *** indicate 10%, 5% and 1% levels of statistical significance, respectively.

**Table 4.** Impact of Textiles on Water PPP Projects (Value Added).

| | (1) | (2) | (3) | (4) | (5) | (6) | (7) | (8) | (9) | (10) |
|---|---|---|---|---|---|---|---|---|---|---|
| | Water Treatment, Sample: All Projects | | | | | Water Treatment: Water Projects Only | | | | |
| **Productivity by Sector and Pollution** | | | | | | | | | | |
| Value-added from textiles in billions of USD | −0.085 *** | −0.086 *** | −0.098 *** | −0.096 *** | −0.093 *** | −0.044 | −0.099 * | −0.096 * | −0.089 * | −0.090 * |
| | (0.016) | (0.019) | (0.021) | (0.021) | (0.021) | (0.033) | (0.039) | (0.040) | (0.041) | (0.041) |
| Proportion of Electricity generated from fossil fuels | −4.360 * | −3.985 | −2.844 | −2.860 | −2.881 | −7.116 | −1.838 | 4.010 | 5.182 | 3.192 |
| | (2.126) | (2.611) | (2.761) | (2.766) | (2.769) | (3.668) | (5.340) | (6.567) | (6.698) | (8.776) |
| Textile value-added × Electricity | 0.113 *** | 0.111 *** | 0.125 *** | 0.123 *** | 0.119 *** | 0.053 | 0.122 * | 0.154 ** | 0.143 ** | 0.142 * |
| | (0.021) | (0.025) | (0.028) | (0.028) | (0.028) | (0.043) | (0.051) | (0.055) | (0.055) | (0.060) |
| **Governance Indicators** | | | | | | | | | | |
| Democracy Score (Polity 2) | | 0.150 | 0.150 | 0.145 | 0.143 | | 0.141 | 0.142 | 0.065 | 0.050 |
| | | (0.077) | (0.077) | (0.078) | (0.078) | | (0.102) | (0.105) | (0.121) | (0.130) |
| Corruption (WGI) | | −0.672 | −0.789 | −0.773 | −0.723 | | −4.941 *** | −5.12 *** | −5.13 *** | −4.622 ** |
| | | (0.695) | (0.711) | (0.713) | (0.715) | | (1.370) | (1.494) | (1.500) | (1.644) |
| **Economic Indicators** | | | | | | | | | | |
| Net FDI inflows in billions of USD (2017) | | | −0.009 | −0.009 | −0.008 | | | −0.028 * | −0.028 * | −0.027 ** |
| | | | (0.006) | (0.006) | (0.006) | | | (0.011) | (0.012) | (0.010) |
| Net exports in billions of USD (2017) | | | 0.001 | 0.001 | 0.001 | | | −0.001 | −0.002 | −0.001 |
| | | | (0.002) | (0.002) | (0.002) | | | (0.003) | (0.003) | (0.003) |
| Poverty HC ratio at $1.90/day (2011 PPP) | | | | −0.010 | −0.009 | | | | −0.044 | −0.038 |
| | | | | (0.019) | (0.019) | | | | (0.041) | (0.044) |
| **Global Demand for Textiles** | | | | | | | | | | |
| Average Cotton price over previous 60 months | | | | | −0.309 * | | | | | −0.203 |
| | | | | | (0.134) | | | | | (0.313) |
| Average Leather price over previous 60 months | | | | | 0.002 | | | | | −0.047 |
| | | | | | (0.016) | | | | | (0.032) |
| Average Synthetic fiber price, previous 60 months | | | | | 0.272 | | | | | 0.482 * |
| | | | | | (0.165) | | | | | (0.233) |
| Chi-squared Statistic | 241.858 | 143.480 | 145.267 | 145.549 | 152.221 | 53.287 | 61.789 | 70.068 | 71.247 | 74.590 |
| N | 6365 | 4835 | 4833 | 4833 | 4833 | 1008 | 870 | 870 | 870 | 870 |
| Textile value-added + [Textiles × Electricity] | 0.028 *** | 0.025 *** | 0.027 ** | 0.027 ** | 0.026 ** | 0.009 | 0.024 ** | 0.057 ** | 0.055 ** | 0.052 * |
| Std. error | (0.005) | (0.006) | (0.012) | (0.012) | (0.012) | (0.010) | (0.012) | (0.023) | (0.023) | (0.028) |
| Country fixed-effects | Yes | Yes | Yes | Yes | Yes | Yes | Yes | Yes | Yes | Yes |
| Year fixed-effects | Yes | Yes | Yes | Yes | Yes | Yes | Yes | Yes | Yes | Yes |
| Sample | All Projects | All Projects | All Projects | All Projects | All Projects | Water PPP | Water PPP | Water PPP | Water PPP | Water PPP |

Notes: Regression analysis of impact of textiles on water PPP Projects (value added). The full sample has 10,427 projects. The symbols *, **, and *** indicate 10%, 5% and 1% levels of statistical significance, respectively.

**Table 5.** Impact of Manufacturing on Water PPP Projects (Value Added).

| | (1) | (2) | (3) | (4) | (5) | (6) | (7) | (8) | (9) | (10) |
|---|---|---|---|---|---|---|---|---|---|---|
| | Water Treatment, Sample: All Projects | | | | | Water Treatment: Water Projects Only | | | | |
| **Productivity by Sector** | | | | | | | | | | |
| Manufacturing value-added in billions of USD | −0.008 *** | −0.008 *** | −0.009 *** | −0.009 *** | −0.009 *** | −0.004 | −0.010 ** | −0.009 ** | −0.009 * | −0.009 ** |
| | (0.001) | (0.002) | (0.002) | (0.002) | (0.002) | (0.003) | (0.003) | (0.004) | (0.004) | (0.004) |
| Proportion of Electricity from fossil fuels | −2.705 | −2.601 | −1.574 | −1.569 | −1.521 | −6.781 | −2.194 | 3.714 | 4.774 | 4.060 |
| | (1.872) | (2.464) | (2.630) | (2.633) | (2.630) | (3.536) | (5.300) | (6.464) | (6.608) | (6.298) |
| Manu. value-added × Electricity from fossil fuels | 0.011 *** | 0.011 *** | 0.012 *** | 0.012 *** | 0.011 *** | 0.005 | 0.012 ** | 0.014 ** | 0.014 ** | 0.014 ** |
| | (0.002) | (0.002) | (0.003) | (0.003) | (0.003) | (0.004) | (0.004) | (0.005) | (0.005) | (0.005) |
| **Governance Indicators** | | | | | | | | | | |
| Democracy Score (Polity 2) | | 0.169 * | 0.167 * | 0.164 * | 0.162 * | | 0.123 | 0.118 | 0.044 | 0.034 |
| | | (0.078) | (0.078) | (0.079) | (0.079) | | (0.100) | (0.102) | (0.118) | (0.115) |
| Corruption (WGI) | | −0.736 | −0.794 | −0.785 | −0.747 | | −5.109 *** | −5.238 *** | −5.250 *** | −5.015 *** |
| | | (0.676) | (0.688) | (0.689) | (0.690) | | (1.367) | (1.469) | (1.479) | (1.406) |
| **Economic Indicators** | | | | | | | | | | |
| Net FDI inflows in billions of USD (2017) | | | −0.008 | −0.008 | −0.007 | | | −0.025 * | −0.025 * | −0.026 ** |
| | | | (0.006) | (0.006) | (0.006) | | | (0.011) | (0.011) | (0.010) |
| Net exports in billions of USD (2017) | | | 0.001 | 0.001 | 0.001 | | | −0.001 | −0.001 | −0.001 |
| | | | (0.002) | (0.002) | (0.002) | | | (0.003) | (0.003) | (0.003) |
| Poverty HC ratio at $1.90/day (2011 PPP) | | | | −0.008 | −0.007 | | | | −0.042 | −0.041 |
| | | | | (0.019) | (0.019) | | | | (0.040) | (0.040) |
| **Global Demand for Textiles** | | | | | | | | | | |
| Average Cotton price over previous 60 months | | | | | −0.287 * | | | | | −0.189 |
| | | | | | (0.133) | | | | | (0.208) |
| Average Leather price over previous 60 months | | | | | 0.003 | | | | | −0.049 |
| | | | | | (0.016) | | | | | (0.037) |
| Average Synthetic fiber price, previous 60 months | | | | | 0.252 | | | | | 0.482 ** |
| | | | | | (0.165) | | | | | (0.164) |
| Chi-squared statistic | 245.990 | 150.954 | 152.354 | 152.515 | 158.289 | 53.949 | 63.659 | 70.203 | 71.328 | 74.670 |
| N | 6653 | 5033 | 5031 | 5031 | 5031 | 1021 | 878 | 878 | 878 | 878 |
| Manu value-added + [Manu × Electricity]-Added | 0.003 *** | 0.002 *** | 0.003 ** | 0.003 ** | 0.003 ** | 0.001 | 0.002 ** | 0.005 ** | 0.005 ** | 0.005 ** |
| Std. error | 0.000 | 0.001 | 0.001 | 0.001 | 0.001 | 0.001 | 0.001 | 0.002 | 0.002 | 0.002 |
| Country fixed-effects | Yes | Yes | Yes | Yes | Yes | Yes | Yes | Yes | Yes | Yes |
| Year fixed-effects | Yes | Yes | Yes | Yes | Yes | Yes | Yes | Yes | Yes | Yes |
| Sample | All Projects | All Projects | All Projects | All Projects | All Projects | Water PPP | Water PPP | Water PPP | Water PPP | Water PPP |

Notes: Regression analysis of impact of textiles on water PPP Projects (value added). The full sample has 10,427 projects. The symbols *, **, and *** indicate 10%, 5% and 1% levels of statistical significance, respectively.

We estimate an impact of several economic and political factors commonly associated with PPPs and FDI on the likelihood that projects will be initiated in the water sector. These include the level of industrial output in textiles and manufacturing, both of which are water-intensive and highly polluting. We also consider the impact a country's reliance on fossil fuels for generating electricity, and the interaction between this reliance and the level of output in the manufacturing and textile industries. The estimates include controls for political and economic factors that are associated with PPPs. Political factors include the levels of democratization and corruption. Economic factors include net foreign direct investment, net exports, the level of poverty, and global demand as reflected in the five-year average prices of cotton, leather, and synthetic fiber.

All regressions are estimated using the xt*logit* command in STATA which is designed for estimating logit models with fixed effects (STATA.com, 2021). All estimates include both country and year fixed-effects. Since our model contains interaction terms, we present the log-odds coefficients—these are best interpreted as associations between the independent and dependent variables. We also present the results of STATA's *lincom* command that calculates the full effect of a variable with an interaction term.

### 3.1. Water Is Different

Scholars have analyzed investment patterns using the PPI data using a variety of techniques but none to date has focused specifically on factors that may affect investment in water utilities or water treatment projects specifically [19]. This project is designed to fill this gap by identifying factors that affect the likelihood that water and water treatment PPPs will be constituted. The results in estimates 1 through 5 in Table 2 suggest that the log odds of water-focused PPPs are inversely correlated with levels of production in the textile

industry (b = −0.34 \*\*\* to b = −0.049 \*\*) relative to all other projects in the PPI database. However, in estimates 1, 2, and 5, the negative coefficients associated with the level of output in the textile industry are significantly offset by their interaction with the proportion of energy the host country generates from fossil fuels. The coefficients for the combined effects of textile value added and the interaction term are reported near the bottom of the table and are positive (b = 0.0042 \* to b = 0.0234 \*\*\*). The magnitude of these coefficients suggests that each billion earned through the textile industry increases the relative odds of a water projects by between 2 percent when no controls are considered, and 0.42 percent when other political and economic factors are considered.

Counter to expectations, however, these results do not persist when we restrict the sample to just PPPs. Estimate 6 shows that the level of production in the textile industry and reliance on fossil fuels for power production do not have a significant impact on the likelihood the PPPs will be created in the water sector. Estimates 7 and 8 suggest, further, that the combination of textile production and reliance on fossil fuels increase the likelihood of PPPs in the electricity sector (b = 0.0162 \*\*), but does not affect the likelihood of PPPs in the transportation sector. We believe that further exploration of these nuances is warranted in future research.

Estimates 1 through 5 in Table 3 suggest that the log odds of water sector projects (b = 170 \*\*\* to b = 0.228 \*\*) increase with production levels in manufacturing. Similarly, water projects are more likely in countries that rely on fossil fuels to generate electricity (b = 4.795 \*\* to 5.067 \*\*). The coefficients for the combined effects of manufacturing value added and the interaction term are also positive and significant in estimates 1, 2, and 4 (b = 1.128 \* to b = 1.469 \*\*). In addition, consistent with our argument that water is different, the combined effects of manufacturing value added and the interaction term remains significant and positive when the likelihood of water PPPs is compared to other PPPs (b = 0.966 \*\*) and it is not significant for electricity or transportation sectors (b = 0.966, b = 1.216).

Variations in the effect of the economic and political control variables add further nuance to our understanding of the uniqueness of the investment in the water sector. Previous research using this dataset for the period 1990–2008 found that political factors and budgetary variables were not strong determinants of PPPs [19]. Our findings bolster these results by confirming that the level of democracy does not have a significant impact on the likelihood of investment in water projects relative to other sectors. At the same time, our findings suggest that the level of corruption may affect investment water projects under certain circumstances. For example, the results in Tables 2 and 3 suggest that PPPs are more likely in the water sector and less likely in the electricity sector when corruption levels are high. As noted above, the WGI indicator of corruption is based on six composite indicators regarding voice and accountability, political stability, government effectiveness, regulatory quality, rule of law, and control of corruption. Combined, these components suggest that higher levels of corruption are likely to be associated with greater level of uncertainty and the resulting risk of loss. It may be that the risk of loss for a given level of uncertainty is lower in water than the electricity sector because of the nature of water was an essential good. This may also explain why the likelihood of water projects and PPPs in the water sector is positively correlated with poverty (b = 0.033 \* to b = 0.043 \*\*, and b = 0.041 \*\* to b = 0.049 \*\* in Tables 2 and 3, respectively), while the opposite is true for PPPs in transportation (b = −0.033 \*\* and b = −0.017 \*\* in Tables 2 and 3, respectively). Water PPPs are also positively correlated with greater net FDI inflows, lower global cotton prices, and higher synthetic fiber prices, while PPPs in the transportation sector are less likely in those circumstances.

To summarize, the likelihood of initiating projects in the water relative to other sectors sector increases when the level of production in manufacturing and textile industries when the country's reliance on fossil fuels for power is considered. These results are mostly consistent with past studies that argue that economic factors such as the size of markets and the level of economic growth are important determinants of PPP placement [19]. Our

analysis, however, takes the additional step of showing that the likelihood of PPPs in the water sector may respond to different combinations of underlying political and economic drivers than other sectors.

### 3.2. Water Treatment versus Water Utilities

Our second hypothesis is that international organizations are likely drawn into funding specific types of water projects. Investments in water utilities have the potential for broader positive spillover effects and direct profit than investments in water treatment. Consequently, while investors may promote water utility projects for a variety of reasons, those who pursue water treatment projects are more likely to have more specific and targeted motivations such as pollution cleanup. At the same time, the political and market characteristics of water utility and water treatment projects make the prospect of public-private partnerships appealing to national governments. As noted above, 99 percent water treatment projects and 95 percent of water utility projects in the PPI database are PPPs. This characteristic allows us to focus on the likelihood of PPPs in water treatment relative to both all other projects in the PPI data base and all other water projects. Consequently, columns 1 through 5 in Tables 4 and 5 estimate the likelihood of the creation of a PPP in water treatment sector relative to the creation of PPPs and non-PPPs in water utilities and all other sectors combined. Columns 6 through 10 in Tables 4 and 5 estimate the likelihood of a PPPs in the water sector will involve water treatment.

Table 4 presents explores the impact of the value-added output of the textile industry on the likelihood of water treatment PPPs. Table 5 explores the impact of the value-added output of the manufacturing sector as a whole. Both sets of results bolster our hypothesis by demonstrating a significant positive relationship between the combined effect of industrial and reliance on fossil fuels for generating electricity (b = 0.035 ** to b = 0.057 ** and b = 0.002 *** to b = 0.005 ** in Tables 4 and 5, respectively). The results reinforce the previous findings that the level of corruption can affect PPPs in the water sector. They show, specifically, that water treatment PPPs are less likely relative to water utility projects when corruption levels are high (b = −5.13 ** to −4.622 ** and b = −5.250 ** to b = −5.015 ** in Tables 4 and 5, respectively). This different may reflect the relative stability and possibility of profit taking even under uncertain political conditions in the water provision compared water treatment and cleanup. PPPs in water treatment are also less likely than PPPs in water utilities when net FDI inflows are low (b = −0.027 ** to −0.028 * and b = −0.026 ** to b = −0.025 ** in Tables 4 and 5, respectively). Neither exports nor the poverty level, cotton prices, or leather prices affect the choice between PPPs in water treatment versus water utility. These results add some nuance to existing literatures that suggest that economic factors such as the size of markets and international exposure are strong predictors of PPP placement [19].

Overall, we interpret these results as evidence that the likelihood of a project being funded as a PPP not only depends on what sector it is in, but also the specific sub-sector. Since water treatment is associated with different political realities than water provision and the number of PPPs appears to decline as net FDI increases there may be an opportunity for international organizations to provide a bridge over troubled waters until corruption problems are reduced and FDI increases.

## 4. Discussion

Given the looming risks of climate change and water scarcity faced by the world, we believe that it is critical to understand the conditions under which investments by private, public, and multilateral actors are likely to take place in the water sector and in other sectors that either consume large amounts of water or generate large amounts of pollution. In the past thirty years, public–private partnerships (PPPs) have been popular mechanisms for encouraging private sector investment and helping local governments overcome economic, political, and technical challenges associated with large infrastructure projects in the water, electricity, and transportation sectors. We demonstrate, however,

that the political economy factors that affect the prevalence of PPPs in the water sector are different than others. Our results confirm, further, that the prevalence of PPPs in the water treatment are distinct from those associated with water utilities even though the delivery of potable water is often negatively affected by water pollution. Most significantly, water treatment PPPs are more likely in countries that have high levels of output in their textile and manufacturing industries and rely heavily on fossil fuels to generate electricity. Furthermore, when corruption levels are high, projects are more likely to be initiated in water than in other sectors, and those investments are more likely to be in water utilities than water treatment. To truly combat the issues of water scarcity, water toxicity, and water security in the years ahead, it will be important to recognize these political economy drivers of investment in the water sector and in those sectors that consume and pollute the most. The efficacy and efficiency of the projects themselves may benefit from a deeper understanding of the drivers that motivate public and private sector investments across different sectors.

**Author Contributions:** Conceptualization, G.S. and S.J.; methodology, G.S. and S.J.; software, G.S. and S.J.; validation, G.S. and S.J.; formal analysis, G.S. and S.J.; investigation, G.S. and S.J.; resources, G.S. and S.J.; data curation, G.S. and S.J.; writing—original draft preparation, G.S. and S.J.; writing—review and editing, G.S. and S.J.; visualization, G.S. and S.J.; supervision, G.S. and S.J.; project administration, G.S. and S.J. All authors have read and agreed to the published version of the manuscript.

**Funding:** This research received no external funding.

**Institutional Review Board Statement:** Not applicable.

**Informed Consent Statement:** Not applicable.

**Data Availability Statement:** All data used in this paper are publicly available. Data on public–private partnerships are drawn from https://ppi.worldbank.org/en/ppi (accessed on 30 July 2021). We used a variety of different measures of corruption and governance. These are as follows: (1) Data on the CPI from Transparency International was downloaded from https://www.transparency.org/en/cpi/2020/index/nzl# (accessed on 30 July 2021). The CPI scores and ranks countries/territories based on how corrupt a country's public sector is perceived to be by experts and business executives. It is a composite index, a combination of 13 surveys and assessments of corruption, collected by a variety of institutions. The composite indicator is measured on a scale of 0–100, where 0 means that a country is perceived as highly corrupt and 100 means that a country is perceived as very clean. (2) The Bayseian Corruption data was downloaded from https://users.ugent.be/~sastanda/BCI/BCI.html (accessed on 30 July 2021). This indicator is widely regarded as an alternative to the Corruption Perception Index (CPI) and the Worldwide Governance Indicators (WGI) published by the World Bank. Methodologically, it is best understood as an augmented version of the Worldwide Governance Indicators' methodology. The underlying source data are aggregated to create an index without any ex-ante imputations and any modeling choices. (3) The International Country Risk Guide data was downloaded via an institutional subscription from Georgetown University. The dataset covers the period 1984—2019. The index ranks 180 countries and territories by their perceived level of public trust. (4) The V-Dem is an index of six indicators of different forms of corruption based on original data from country experts. The dataset covers the period 1900—2012. Multiple coders, at least three-fifths of whom are native to the countries in question, were used to code each country-year observation, and the coder recruitment procedures and coding procedures were consistent over time and across countries.

**Acknowledgments:** We appreciate excellent research assistance from Joshua Levy, Victor Li, Jack Little, Evan Laugen, and Karan Pratap Chauhan in this paper.

**Conflicts of Interest:** The authors declare no conflict of interest.

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
