# Peer review of "Bridges over Troubled Waters? The Political Economy of Public-Private Partnerships in the Water Sector"

_sustainability, doi:10.3390/su131810127_

Round 1

Reviewer 1 Report

It is a nice paper.  The only revision I am seeking is to put a better face and use more context about the variables.  It is OK to say, for example, that the parameter on corruption is negative so more corruption means fewer PPP.  Corruption means different things in different places.  More context would really make your results pop off the page and be more compelling.  I am not asking for a lot and leave the choices of what to highlight and provide more depth to to the researchers.  

Author Response

Thank you for your comments and suggestions.  We have added more discussion of corruption and other influential factors throughout. Since we have made some changes to the empirical approach (due to comments from another referee), the results section is different than the previous draft. But we have taken this comment to heart, added more context, more comparisons of our results to others in the literature and more suggestions for future research. We hope this makes the paper more interesting and readable!

Reviewer 2 Report

This manuscript studies the drivers for PPPs in the water sector, and especially whether there exist any differences in the funding of PPPs between water treatment projects and water sanitation projects. I find the research question fascinating, the theories well-motivated, and the mechanisms well-explained. The manuscript is well written and it is a delight to read through the passage.

My major concerns are with the econometric design of the paper. While I believe the empirics are potentially fixable at this point, major changes are needed.

Major Comments:

  • In the database, you observe the universe of PPPs, and you estimate a linear probability model (LPM) on the probability of a given PPP project belongs to a certain category, and implicitly, not other categories.
  1. Given that these probabilities actually sum up to one, you should be modeling these outcomes using multinomial logit (MNL) / multinomial probit (MNP) rather than separate LPM equations, which does not correctly capture the correlation between outcomes.
  2. Taking the models as given, these LPMs point to the intensive margin of the projects, i.e., given an implemented PPP, is it more likely to be a sanitation or a water treatment project. What your two hypotheses claim, on the other hand, is the extensive margin effect of whether water treatment PPPs are more/less likely to happen given your independent variables. In other words, you would want to test the hypothesis of whether the covariates drive the total numbers of water treatment projects, not whether they lead to more water treatment projects than water sanitation ones. I don’t think this is what LPM, or for that matter, MNL/MNP, provides you with.
  • Similar problem with equations on the likelihood of PPP projects (columns 1-5).
  1. You would want to do a logit/probit, which correctly models the binary structure of the outcome variable. LPM could probably be justified in this case as there are only two potential outcomes, but at least a robustness check is necessary.
  2. These LPMs point to the likelihood of a project funded by PPP over the omitted baseline. First, please explain the nature of those non-PPP projects in the database. Are they not funded at all, or are they funded through some other means? Second, assuming that the non-PPP projects are also funded, these models essentially point to the likelihood of projects funded by PPP over other means of funding, and NOT the actual amount of PPPs. I think the latter is what you are really after here with the way the hypotheses are set up.
  • Do you observe the size of each project, potentially through the amount funded? Econometrically, this should at least be a control variable if you observe this. Practically, your model setup essentially treats larger projects and smaller projects as equally important. What are the implications of this assumption?

Minor Comments:

Are country fixed-effects included in the models? Line 250 suggests only year FEs are included. Table notes suggest otherwise. I would be concerned with potential omitted variable bias if country FEs are not included.

Line 41: the use (of) public private partnerships

Line 126: PPPs are (more) likely to be involved in water provision

Author Response

Responses – we have placed our responses in italics

Referee #2

This manuscript studies the drivers for PPPs in the water sector, and especially whether there exist any differences in the likelihood of funding of PPPs between water treatment projects and water sanitation projects. I find the research question fascinating, the theories well-motivated, and the mechanisms well-explained. The manuscript is well written and it is a delight to read through the passage.

We appreciate the kind words! Thank you!

My major concerns are with the econometric design of the paper. While I believe the empirics are potentially fixable at this point, major changes are needed.

We altered the econometric design in response to this concern.  We did so using logit estimations of the probability that a PPP in the water sector or water treatment would be created at a particular time, controlling for country-specific and year-specific fixed effects.  This technique allows us to focus specifically on the likelihood of water and water treatment projects.  We agree with the reviewer that multinomial logistic regression would provide additional information about comparisons across the water, power, and transportation sectors.  We also agree that analyzing the amount of funding would provide additional insights.  Both are complex and nuanced given the project-level nature of the data and variations in the degree of funding throughout individual project life cycles.  Given the limited time we were given to address these changes, we chose to keep our focus narrowly on the likelihood that water or water treatment PPPs would funded and recommend these analyses as fruitful avenues for future research. 

Major Comments:

In the database, you observe the universe of PPPs, and you estimate a linear probability model (LPM) on the probability of a given PPP project belongs to a certain category, and implicitly, not other categories.

  1. Given that these probabilities actually sum up to one, you should be modeling these outcomes using multinomial logit (MNL) / multinomial probit (MNP) rather than separate LPM equations, which does not correctly capture the correlation between outcomes.

This is an insightful point and we fully agree with it. We have removed the LPM. To estimate a multinomial logit/probit model with country and year fixed-effects however, we need a lot of time. Estimating a single equation is taking several hours – we are unable to complete the analysis in the tight timeline for this special issue. So instead, we have acknowledged this as an important topic of future research and focused our analysis on the funding of water projects. We have de-emphasized the decision about other types of projects. All our analysis relies on the logit model with country and year fixed-effects.

  1. Taking the models as given, these LPMs point to the intensive margin of the projects, i.e., given an implemented PPP, is it more likely to be a sanitation or a water treatment project. What your two hypotheses claim, on the other hand, is the extensive margin effect of whether water treatment PPPs are more/less likely to happen given your independent variables. In other words, you would want to test the hypothesis of whether the covariates drive the total numbers of water treatment projects, not whether they lead to more water treatment projects than water sanitation ones. I don’t think this is what LPM, or for that matter, MNL/MNP, provides you with.

Similar problem with equations on the likelihood of PPP projects (columns 1-5).

  1. You would want to do a logit/probit, which correctly models the binary structure of the outcome variable. LPM could probably be justified in this case as there are only two potential outcomes, but at least a robustness check is necessary.

Indeed, we have removed the LPM and rely instead of a logit model which corrects for the binary structure of the outcome variable. Our results are actually much stronger, so it is the LPM that should be regarded as a robustness check.

  1. These LPMs point to the likelihood of a project funded by PPP over the omitted baseline. First, please explain the nature of those non-PPP projects in the database. Are they not funded at all, or are they funded through some other means? Second, assuming that the non-PPP projects are also funded, these models essentially point to the likelihood of projects funded by PPP over other means of funding, and NOT the actual amount of PPPs. I think the latter is what you are really after here with the way the hypotheses are set up.

We have addressed this issue in the current version of the paper. On page 4, we add the following paragraph: “Our analysis focuses on the probability of developing PPPs in different sectors over time. The official database currently provides information on 10,427 infrastructure projects dating from 1984 to 2020 and is updated with last year’s data six months after year-end. It contains over 50 fields per project record, including country, financial closure year, infrastructure services provided, type of private participation, technology, capacity, project location, private sponsors, private debt providers and development bank support.  The PPI dataset includes 5,898 projects in the power sector (57 percent), 3,017 projects in the transportation sector (29 percent), and 1,512 in the water sector (14 percent).  Of those, 498 are water utility projects (33 percent) and 1, 014 are water treatment projects (67 percent).  8,709 (84 percent) of the projects in the data set are PPPs. Almost all water treatment projects are PPPs (675 out of 682, 99 percent), and almost all water utilities are PPPs (472 out of 498, 95 percent).”

We also note here that the list of non-PPP projects is not exhaustive. It represents the best efforts of a research team at World Bank to compile publicly available information on large infrastructure projects (PPP and single-source funded). Some projects — particularly those involving local and small-scale operators — tend to be omitted because they are usually not reported by major news sources, data-bases, government websites, and other sources used the PPI Projects database. Large projects however, that are comparable to PPPs, are well-represented for each of the countries.

We have also clarified the samples we are working with in the tables as well as the writeup. Tables 2 and 3 for example, present the results for all projects in the database as well as the restricted sample of just PPP projects. In our writeup, we are now much clearer about the reference group.

Do you observe the size of each project, potentially through the amount funded? Econometrically, this should at least be a control variable if you observe this. Practically, your model set up essentially treats larger projects and smaller projects as equally important. What are the implications of this assumption?

We do have some information about the sizes of projects, but we refrain from adding this as a control variable for two reasons. Firstly, we only constituently observe the size of the project in the year of funding.  This is problematic because our previous research suggests that PPPs are notorious for going vastly over-budget and so the measurement error is likely to be quite significant.  Reporting of additional funding over the life of the project and the length of the life of the projects vary.  Second, the project size variable is not inflation-adjusted and for many countries in our sample, we do not have price deflators for all the years in the sample. Finally, we refrain from modeling the size of the project because we deliberately want to treat large and small projects on a level playing field. Almost all road-building projects tend to be substantially larger than a water-treatment plant or a water-utility program. For this project, we are most interested in whether a water treatment or utility program was funded at all, not the scale of the project. We agree that exploring the nuances of financing volume and type are fruitful areas for future research.

Minor Comments:

Are country fixed-effects included in the models? Line 250 suggests only year FEs are included. Table notes suggest otherwise. I would be concerned with potential omitted variable bias if country FEs are not included.

Country and year fixed-effects are both included.  They were also included in the last version of the paper. We apologize that this was not clear.

Line 41: the use (of) public private partnerships

We have corrected this error.

Line 126: PPPs are (more) likely to be involved in water provision

Indeed, this was an error. We have fixed the issue. Apologies for the poor use of terms!

Round 2

Reviewer 2 Report

Thank you for the reply. I am really impressed by the amount of work done in such a short period. I also agree that MNL with individual FEs is not the friendliest model to deal with given time constraints, and this is NOT an economics journal. 

I look forward to seeing this paper in print.